# Hydroxytyrosol Reduces Foam Cell Formation and Endothelial Inflammation Regulating the PPARγ/LXRα/ABCA1 Pathway

**DOI:** 10.3390/ijms24032057

**Published:** 2023-01-20

**Authors:** Sara Franceschelli, Federica De Cecco, Mirko Pesce, Patrizio Ripari, Maria Teresa Guagnano, Arturo Bravo Nuevo, Alfredo Grilli, Silvia Sancilio, Lorenza Speranza

**Affiliations:** 1Department of Medicine and Aging Sciences, University “G. d’Annunzio” Chieti-Pescara, Via dei Vestini 31, 66100 Chieti, Italy; 2Department of Innovative Technologies in Medicine & Dentistry, University “G. d’Annunzio” Chieti-Pescara, Via dei Vestini 31, 66100 Chieti, Italy; 3Department of Bio-Medical Sciences–PCOM, Philadelphia College of Osteopathic Medicine, Philadelphia, PA 19131, USA

**Keywords:** cholesterol, foam cells, oxLDL, hydroxytyrosol, adhesion molecules, inflammation, macrophage

## Abstract

Cholesterol accumulation in macrophages leads to the formation of foam cells and increases the risk of developing atherosclerosis. We have verified whether hydroxytyrosol (HT), a phenolic compound with anti-inflammatory and antioxidant properties, can reduce the cholesterol build up in THP-1 macrophage-derived foam cells. We have also investigated the potential mechanisms. Oil Red O staining and high-performance liquid chromatography (HPLC) assays were utilized to detect cellular lipid accumulation and cholesterol content, respectively, in THP-1 macrophages foam cells treated with HT. The impact of HT on cholesterol metabolism-related molecules (SR-A1, CD36, LOX-1, ABCA1, ABCG1, PPARγ and LRX-α) in foam cells was assessed using real-time PCR (RT-qPCR) and Western blot analyses. Finally, the effect of HT on the adhesion of THP-1 monocytes to human vascular endothelial cells (HUVEC) was analyzed to study endothelial activation. We found that HT activates the PPARγ/LXRα pathway to upregulate ABCA1 expression, reducing cholesterol accumulation in foam cells. Moreover, HT significantly inhibited monocyte adhesion and reduced the levels of adhesion factors (ICAM-1 and VCAM-1) and pro-inflammatory factors (IL-6 and TNF-α) in LPS-induced endothelial cells. Taken together, our findings suggest that HT, with its ability to interfere with the import and export of cholesterol, could represent a new therapeutic strategy for the treatment of atherosclerotic disease.

## 1. Introduction

Atherosclerosis (AS) is a chronic inflammatory vessel disease triggered by multiple environmental and genetic factors. Macrophage foam cell formation and endothelial cell inflammation and activation can lead to the beginning, as well as the development, of AS, mediated by lipid and cholesterol accumulation within the walls of large and medium arteries [1]. Increased levels of plasma, low density lipoprotein (LDL) and cholesterol, and their retention and accumulation in the sub-endothelial intima, are the main causes of the chronic inflammatory response. As a result, the activated endothelium releases chemokines, such as monocyte chemotactic protein-1 (MCP-1), as well as inflammatory cytokines, such as tumor necrosis factor-α (TNF-α) and interleukin-1β (IL-1β), which in turn promote local inflammation [2]. Moreover, the expression of adhesion molecules on the endothelial surface, including the P- and E-selectins, the intercellular adhesion molecule-1 (ICAM-1) and the vascular cell adhesion molecule-1 (VCAM-1), are responsible for the adhesion of circulating monocytes to the vessel wall [3]. The monocytes differentiate into macrophages, after migrating below the endothelium, contributing to the immune response against trapped LDL [4]. In the intima, the exposition to oxidant agents determines the LDL conversion into oxidized forms (oxLDL) which can result in atherosclerotic plaque formation. In fact, oxLDL reduces cholesterol efflux from macrophages, leading to the development of foam cells [5]. The oxLDL are phagocytized into macrophages due to how they bind to scavenger receptors, including Class A1 scavenger receptors (SR-A1), cluster of differentiation 36 (CD36) and lectin-like ox-LDL receptor-1 (LOX-1), in turn stimulating the cellular accumulation of cholesterol. oxLDLs are degraded to oxysterols, which are inflammatory components responsible for the formation of atherosclerotic plaque [6]. The balance between intracellular hydrolysis and esterification of cholesterol plays a pivotal role in regulating cholesterol homoeostasis and the prevention of foam cell formation. The deposited cholesterol can be removed from macrophages using reverse transport mediated by ATP-binding cassette (ABC) transporters, such ABC transporter A1 (ABCA1) and transporter G1 (ABCG1), leading to the formation of high-density lipoprotein (HDL) from macrophages [7]. In mice, the deficiency of ABCA1 and ABCG1 has been shown to trigger the buildup of excess cholesterol in macrophages, inducing the formation of atherosclerotic plaque [8]. Peroxisome proliferator-activated receptor-γ (PPARγ) and liver X receptor α (LXRα) are nuclear receptors involved in the regulation of lipid and glucose metabolism and are activated by oxidized fatty acid ligands and oxysterols, respectively. These nuclear receptors therefore act as intracellular cholesterol sensors inducing ABCA1 and ABCG1 expression and removing the cholesterol excess [9]. Since PPARγ and LXRα play a key role in reverse cholesterol transport (RCT), their activation could lead to a reduction in the formation of foam cells and thus have a negative effect on atherogenesis reducing cardiovascular disease risk. Numerous studies have demonstrated the importance of naturally occurring dietary polyphenols in promoting cardiovascular health. Multiple natural products (flavonoids, terpenoids, phenolic compounds, phenylpropanoids, alkaloids, steroids, fatty acids, amino acids, as well as carbohydrates) can inhibit foam cell formation and thus exhibit anti-atherosclerotic capacity by suppressing lipid uptake, cholesterol esterification and/or promoting cholesterol ester hydrolysis and cholesterol efflux [10]. The inhibition of foam cell deposition within the aortic intima and the formation of atheroma can be prevented or reduced by the atheroprotective action of various natural molecules, such as berberine, resveratrol, hibiscus, allicin, lenourine, Terminalia arjuna, Salvia miltiorrhiza, Pueraria lobata, as well as Ginkgo biloba [11,12,13,14,15,16,17,18].

The Mediterranean diet represents a healthy and complete food model from a nutritional point of view. This diet is based on a consistent consumption of foods that are rich in healthy nutrients, such as fish, fruit, vegetables, legumes and cereals, and on a low consumption of foods rich in proteins (for example red meats) and saturated fats. In the Mediterranean diet, olive oil, especially extra virgin olive oil (EVO), is the main source of vegetable fats [19]. Unlike other vegetable oils, olive oil contains high quantities of numerous active ingredients, including oleic acid (which represents the predominant monounsaturated fatty acid) and polyphenols (hydroxytyrosol, tyrosol, oleuropein and flavonoids) responsible for organoleptic characteristics that distinguish olive oil (for example its characteristic bitter taste), phytosterols, tocopherol (vitamin E) and squalene. Recently, natural compounds have been the focus of interest in several studies for atherogenesis prevention. In vitro and in vivo studies have described health promoting actions associated with their anti-inflammatory, antioxidant, antithrombotic and antiatherogenic effects [20,21,22]. Several studies have established the ability of polyphenols to inhibit atherosclerotic progression. Various mechanisms have been proposed to elucidate the anti-atherosclerotic properties of polyphenols comprising, among others, inhibition of LDL oxidation, modulation of platelet aggregation, diminution of inflammation in macrophages and modulation of cholesterol levels. Polyphenols also seem to have the ability to influence LDL uptake, interfering with the LXRα receptor, as many studies have reported a reduction in oxidized LDL accumulation in macrophages [23,24].

For this reason, it is necessary to improve the knowledge of biological mechanisms to elucidate the health effects of bioactive molecules present in natural products. Hydroxytyrosol (HT) is a natural bioactive phenolic compound present in high concentrations in olives and virgin olive oil, and is well known for its anti-microbial, anti-inflammatory, antioxidant and free radical-scavenging properties [25,26]. It originates from the hydrolysis of oleuropein by an esterase during the milling process. However, this valuable substance is not commercially available, since the various production methods used to obtain large quantities of HT are very expensive and sometimes use toxic reagents. HT is also biosynthesized from tyrosine. Studies have shown that HT can be produced directly by an intermediate of this metabolism, tyrosol, by bioconversion using whole cells or enzymes as biocatalysts. In recent years, the biosynthesis of tyrosol has received increasing attention because of its low cost, high production efficiency and environmental friendliness. Tyrosol can be produced by microbial fermentation and enzymatic conversion. For example, yeast synthesizes tyrosol from tyrosine via the Ehrlich pathway, through transamination, decarboxylation and reduction by aromatic amino acid transferase, pyruvate decarboxylase and alcohol dehydrogenase, then is hydroxylated to HT [27]. HT is a small and hydrophilic molecule, which means that it is easily absorbed and subsequently excreted in the urine as a glucuronide conjugate [28]. In recent years, it has been well documented that this phenolic compound has various health benefits, and it has been shown in preclinical studies to have protective actions against several diseases.

Previous studies reported that HT reduced vascular inflammation, NF-κB activity, the expression of endothelial adhesion molecules (VCAM-1, ICAM-1 and E-selectin) and production of reactive oxygen species (ROS) [29,30,31]. The antioxidant properties of HT negatively influence the oxidation of LDL, leading to a reduction in the formation of atherosclerotic plaques [32,33]. Furthermore, extracts doped with HT decreased the cardiovascular risk through the reduction in plasma levels of cholesterol, lipids [34] and blood pressure [35]. In cholesterol-fed rats, HT treatment induced an increase in HDL and a decrease in serum LDL, but the mechanisms that explain this antiatherogenic property of HT have yet to be elucidated [34]. A meta-analysis of 18 studies (cohort of 4,172,412 subjects) showed that the consumption of the Mediterranean diet resulted in an 8% reduction in overall mortality. These studies also demonstrated a 10% decrease in the risk of developing cardiovascular disease [36]. Considering the findings mentioned above, in this study we will explore the possible role of HT in regulating cholesterol homoeostasis through the analysis of the main molecular pathways involved in PPARγ/LXRα signaling.

## 2. Results

### 2.1. HT Does Not Affect the Viability and Mitigates Oxidative Stress in THP-1 Macrophage Foam Cells

To determine whether HT affects cell viability of THP-1 macrophage-derived foam cells, an MTT assay was performed. As shown in Figure 1a, HT (from 25 µM to 2 mM) did not reduce cell viability at 24 h and 48 h. Given the antioxidant mechanisms of HT, we focused on studying the influence of one of the ROS in HT-treated foam cells. As expected, oxLDL increased ROS production. Treatment with HT concentrations above 50 µM significantly reduced (*p* < 0.01) O_2_^−^ generation compared to untreated foam cells (Figure 1b). These results suggest that HT effectively influences the production of ROS in foam cells. For subsequent experiments, we chose concentrations of 25, 50 and 100 µM of HT.

### 2.2. HT Decreases Lipid Over-Accumulation in THP-1 Macrophage Foam Cells

Lipid accumulation in macrophages and the formation of foam cells play a crucial role in all stages of the development of the atherosclerotic lesion [37,38,39]. We used PMA to induce monocyte THP-1 differentiation into macrophages. Next, ox-LDLs were employed to stimulate transformation of THP-1 macrophages into foam cells. The Oil Red O staining assay is crucial to examining foam cell formation and to detect lipid droplets distributed in the cytosol. As shown in Figure 2a, few lipid droplets were found in the untreated macrophages. THP-1 macrophages assumed the morphological appearance of a foam cell after incubation with oxLDL for 24 h, with ORO-stained lipid droplets present throughout the cytosol of most cells. The first objective of our study was to evaluate whether HT can regulate lipid accumulation in THP-1 macrophage foam cells. As shown in Figure 2a, a microscopic examination of foam cells treated with 25, 50 and 100 µM of HT for 24 h showed that HT was able to reduce intracellular red-stained particle accumulation in foam cells at concentrations greater than 50 µM. In particular, the quantitative data of ORO staining indicated that treatment with 50 μM HT led to a decrease in lipid accumulation. Thus, this concentration was chosen to perform the subsequent experiments to study the mechanism of HT against lipid droplet accumulation. To support these results, we also evaluated the total intracellular cholesterol level in foam cells. In ox-LDL-induced THP-1 macrophages, the total cholesterol (TC) level and free cholesterol (FC) were significantly increased when compared to macrophage cells (*p* < 0.01, Figure 2b). This result represents further confirmation that we have successfully obtained a foam cell model. Furthermore, intracellular TC and cholesterol ester (CE) levels and the CE/TC rate were markedly reduced by treatment with 50 µM HT (Figure 2b,c) when compared to untreated foam cells.

### 2.3. Effect of HT on Cholesterol Metabolism-Related Molecules

In THP-1 macrophage foam cells treated with HT, we evaluated the mRNA levels of many cholesterol metabolism-related genes (SR-A1, CD36, LOX-1, ABCA1, ABCG1 and LRX-α) by qPCR. An analysis of scavenger receptors showed that HT reduced CD36 mRNA expression compared to untreated foam cells but had no significant effect on the expression of the LOX-1 and SR-A1 genes. In addition, HT significantly increased the mRNA levels of LXRα and ABCA1, but no modulations in ABCG1 were observed as compared to untreated foam cells (Figure 3).

Finally, Western blot analysis of CD36, LRX-α and ABCA1 confirmed our results (Figure 4) indicating that the effect of HT on cholesterol efflux from foam cells is probably due to the regulation of scavenger receptor CD36 and LXRα/ABCA1 signaling. Notably, we did not observe the same correlation between mRNA levels and protein expression in macrophages, but as extensively discussed by Vogel et al., this could be due to post-transcriptional, translational and protein degradation regulatory mechanisms [40].

### 2.4. Effect of HT on PPARγ/LXRα/ABCA1 Signaling in Human THP-1 Macrophage Foam Cells

The transcriptional cascade in the PPARγ/LXRα pathway plays a crucial role in maintaining cellular cholesterol homeostasis in macrophages. Many studies have shown that the activation of PPARγ upregulates ABCA1 expression and improves cellular cholesterol efflux [41,42]. Thus, we investigated whether HT treatment could induce PPARγ mRNA and protein expression modulation in foam cells. As showed in Figure 5, the HT treatment upregulated PPARγ expression with respect to untreated foam cells.

Since the ATP-binding cassette transporter ABCA1 is elicited following LXRα activation by PPARγ agonists [43], the effects of HT on the LXRα and ABCA1 protein expression levels were evaluated in foam cells pre-treated with PPARγ antagonist GW9662 or LXRα antagonist GSK2033, followed by analysis of the molecules downstream. GW9662 and GSK2033 prevented the induction of ABCA1 promoted by HT (Figure 6a).

In addition, in cells treated with the PPARγ antagonist, LXRα upregulation was also abolished (Figure 7). These findings demonstrate that HT, at 50 µM, behaves as an agonist for PPARγ in the context of foam cells, promoting cholesterol efflux via the PPARγ/LXRα/ABCA1 pathway.

### 2.5. HT Decrease LPS-Induced Adhesion and Inflammatory Response in Endothelial Cells

As PPAR-γ is an important negative regulator in LPS-induced inflammatory responses [43], we monitored the effects of HT by evaluating the monocyte adhesion to inflamed endothelial cells. A small number of monocytes were adhered to unstimulated HUVEC monolayers, but strongly adhered to LPS-activated endothelial cells (Figure 8a). This effect was found to be reduced when activated HUVEC cells were exposed to 50 µM HT. This result was confirmed by the analysis of the mRNA levels of the adhesion molecules ICAM-1 and VCAM-1 which, induced by LPS, are inhibited by treatment with HT (Figure 8b). The qPCR analysis showed that the levels of LPS-induced pro-inflammatory cytokines, TNF-α and IL-1β, were also strongly reduced by treatment with HT (Figure 8c). Finally, to better support the observed anti-inflammatory activity of HT, the level of superoxide anion was further analyzed in this cellular model as well. As shown in Figure 8d, the superoxide anion, a marker of oxidative stress, was significantly higher in HUVEC-activated cells, while a dose of 50 µM HT reduced its levels. Overall, this data shows that HT protects against oxidation-driven and inflammation-associated endothelial dysfunction through the reduction in pathways related to activation of adhesion molecules.

## 3. Discussion

Scientific studies have shown that phenols such as hydroxytyrosol (HT), a bioactive component contained in olive extracts, exert a wide range of biological effects, i.e., cardioprotective, anticancer, antimicrobial and neuroprotective [44,45,46,47]. The exact molecular mechanisms underlying many of these actions, such as the effects of HT on lipid accumulation in foam cells, have yet to be fully clarified. Based upon the fundamental role of the lipid metabolism and inflammatory response in the initiation of the atherosclerotic process, the current study evaluated the effects of HT on the lipid accumulation in foam cells and cholesterol efflux. This could represent a new line of research that will be able to broaden the knowledge on the dualism between antioxidant principles extracted from natural compounds and prevention of some diseases characterized by excess uptake of ox-LDL, foam cell formation or impaired cholesterol efflux in macrophages [48]. Many studies have focused on the effect of polyphenols (such as curcumin and quercetin) on the metabolic changes in cholesterol metabolism via LXRα. These results, even without having explicitly explained a mechanism of action, have nevertheless revealed the relationship between polyphenols and LXR α receptors [14,40,49]. Considering this experimental evidence, we have hypothesized that HT could regulate LXR pathways, and we used ox-LDL-induced foam cells to demonstrate that HT influences cholesterol efflux by regulating the PPAR-γ/LXRα/ABCA1 pathway.

In our experiments, Oil Red O (ORO) staining of lipid-laden foam cells was used to discriminate the appropriate concentration of HT capable of limiting the formation of lipid droplets. Our data evidenced that a concentration of 50 µM of HT significantly reduced lipid droplet formation compared to untreated foam cells.

Recent studies have shown that ox-LDL has pro-inflammatory and pro-oxidative stress effects, inducing the inflammation of vascular walls, thus becoming a determining factor in the development of plaques [50]. The redox system allows cells to respond, through modifications of energy and/or informational flows mediated by electronic transfers, to a series of exogenous or endogenous “stressors” to maintain the homeostasis of the whole organism. Of course, any alteration—congenital or acquired—at the expense of this complex system can cause, through an inadequate response to the various "stressors", a series of molecular alterations on an oxidative basis which, if not promptly and effectively corrected, can alter the energy flows. This could damage the cells and therefore predispose the individual to pathologies [51]. Mitochondrial enzymes produce O_2_^−^ at physiological levels; however, exogenous and endogenous pathological mechanisms can alter mitochondrial function and lead to pathological levels of ROS, thus triggering an oxidative process [50]. Our results indicated that HT inhibits ox-LDL-induced ROS generation. Therefore, HT protects macrophages against ox-LDL-induced oxidative damage, at least in part, by inhibiting ROS production. This is relevant considering that ROS induce the expression of scavenger receptors (SRs) in macrophages (including SRA1, CD36 and LOX-1) involved in oxLDL uptake, and as a result are overregulated in the early stages of AS. To explore the mechanism in which HT prevents the formation of foam cells, we assessed the expression of SRs in foam cells. In the present study, we found that treatment with 50 µM HT induces downregulation of CD36 expression but does not influence the expression of the LOX-1 and SRA-1 receptors. This is probably related to the fact that the internalization of oxLDL by monocytes/macrophages occurs predominantly through the class B scavenger receptor, CD36, and to a lesser extent through other SRs [52,53]. Downregulation of CD36 expression leads to a decrease in oxLDL absorption and to an accumulation of lipid droplets. This should lead to a reduced formation of macrophage foam cells, thus becoming one of the determining factors in the triggering of the atherosclerotic process. The LXRs are nuclear receptors that bind oxysterols and activate a set of gene encoding proteins involved in the reverse transport of cholesterol from peripheral tissues to the liver [7]. LXR agonists promote reverse cholesterol transport via ABCA1 and ABCG1, protecting against atherosclerosis. A deficiency of LXRs induces an accumulation of sterols in tissues and an accelerated atherosclerosis process [8]. Our results showed that in foam cells, HT treatment increased cholesterol efflux by upregulating LXRα expression. This was confirmed by the increase in the expression of the membrane transporter expression ABCA-1, which mediates cholesterol efflux and is critically involved in removing the excess cell cholesterol from foam cells. Since the literature shows that PPARs and LXRs have a well-defined role in regulating lipid homeostasis and in metabolic diseases [9], we explored the possible role of HT in the expression of these nuclear transcription factors. Natural ligands for PPARγ are polyunsaturated fatty acids and eicosanoid acids. Several studies have reported that some polyphenolic molecules act as a PPAR-γ agonist, controlling lipidic metabolism [54]. In the present study, we showed that HT upregulates PPAR-γ expression in THP-1 macrophage foam cells, and this increase was abolished by the specific PPARγ antagonist GW6992, evidencing that HT promotes ABCA1 expression through the regulation of PPARγ/LXRα signalling, whose interactions are essential for cholesterol homeostasis. These results from the first part of our study led us to hypothesize that HT may have a determining role in the prevention of atherogenic plaque formation. We have demonstrated its ability to inhibit molecular mechanisms essential for its development. As mentioned above, in addition to cholesterol metabolism disorder, inflammation is also considered a major risk factor for AS. Inflammation encourages macrophage uptake of lipids and foam cell formation, playing a key role in endothelial dysfunction. Increased secretion of inflammatory cytokines such as IL-1β and TNF-α because of endothelial injury can modulate the expression of many genes involved in the process of AS [55,56]. In this context, adhesion molecules such as ICAM-1 and VCAM-1 encourage inflammation through mediating monocyte adhesion to the endothelial cells and turning them into foam cells [3,56]. In this research, we monitored the effects of HT on monocyte adhesion to inflamed HUVEC endothelial cells. We found that HT inhibited monocyte adhesion and inflammatory responses, decreasing the expression of adhesion molecules, as shown in Figure 8b. Furthermore, the reduction in the levels of inflammatory markers such as IL-1β and TNF-α (Figure 8c) is an additional demonstration of the antioxidant effect of HT. In fact, our data confirm that HT, in a cellular model of endothelial dysfunction, reduces ROS (Figure 8d), which represents a determining factor for the triggering of endothelial dysfunction.

## 4. Materials and Methods

### 4.1. Cell Culture

The human monocytic leukemia THP-1 cell line (ATCC^®^ TIB-202™ Rockville, MD, USA) was cultured in a 5% CO_2_ atmosphere at 37 °C in RPMI medium containing 10% fetal bovine serum (FBS), 100 ng/mL streptomycin, 100 U/mL penicillin and 2 mM L-glutamine (Sigma-Aldrich, St. Louis, MO, USA). Differentiated adherent THP-1-derived macrophages were obtained by seeding 1 × 10^6^ cells and treating them with 100 nM phorbol 12-myristate 13-acetate (PMA, Sigma-Aldrich, P8139) for 48 h. Expression of the CD14 molecule in THP-1 cells and differentiated macrophages was detected using the mouse monoclonal antibody against human CD14 ELISA to confirm cell differentiation. Before stimulation with hydroxytyrosol (HT) (N20100141, Natac, Madrid, Spain) at the indicated concentration for 24 h, cells were pre-incubated with 50 μg/mL oxLDL (Invitrogen, Waltham, MA, USA) for 24 h to induce the formation of foam cells. The specific peroxisome proliferator-activated receptor-γ (PPAR-γ) antagonist, GW9662 (10 μM, Sigma-Aldrich), or the LXRα inhibitor, GSK2033, (1 μM, Sigma-Aldrich) was added 30 min before HT treatment. An ethanolic solution of HT was reconstituted in culture medium to make a 1 mM stock solution (ethanol concentration in 1 mM stock was 0.3%, *v*/*v*) and was further diluted for experimental purposes. Controls were treated with an equal ethanol concentration.

### 4.2. Monocyte-HUVEC Adhesion

The human umbilical vein/vascular endothelium cell line (HUVEC) (ATCC^®^ CRL-1730™) was cultured in a 5% CO_2_ atmosphere at 37 °C in Endothelial Cell Growth Medium (Cell Applications, Inc, San Diego, CA, USA). HUVECs were then seeded in duplicate in a 24-well plate on a sterile glass coverslip at a density of 5 × 10^4^ in 400µL of endothelial cell medium. HUVECs grown to confluence were incubated with 1µg/mL of lipopolysaccharides from Escherichia coli (LPS, Sigma Aldrich) and/or 50 µM of HT for 6 h. After medium removal, the cell monolayer was washed twice with PBS. At this point, THP-1 cells (2.5 × 10^5^) were seeded over the monolayer at ratios of 1:5 with respect to the HUVECs in complete RPMI medium. After 1 h of incubation, supernatants were aspirated and HUVEC monolayers plus attached THP-1 cells were washed twice with PBS. Cellular pellets were collected for RNA and protein extraction.

### 4.3. Cytotoxicity Assay and ROS Detection

A methylthiazolyldiphenyl-tetrazolium bromide (MTT) assay (Sigma-Aldrich, St. Louis, MO, USA) was used to assess the metabolic activity of THP-1 macrophage foam cells treated with different concentrations of HT (25, 50, 100, 250, 500, 1000 and 2000 µM). As reported previously, foam cells were seeded in duplicate at 5 × 10^5^ in 100 µL of complete medium in a 96-well plate [57]. Cells were maintained in fresh serum-free medium for 4 h to synchronize their growth. Afterwards, the medium was replaced with fresh serum-free medium containing various concentrations of HT, and the cells were then incubated for 24 h and 48 h. MTT reagent was added to each well. After incubation at 37 °C for 2 h, the yellowish solution was converted to intracellular, dark blue, water-insoluble MTT formazan crystals by mitochondrial dehydrogenases of living cells. The MTT/medium was removed and replaced with an equal volume of DMSO (Sigma-Aldrich, St. Louis, MO, USA) to dissolve cells and the intracellular formazan crystals. The absorbance of the resulting purple solution was measured at 570 nm on a spectrometer. The same procedure was used to evaluate HUVEC cell viability treated with 50 µM HT which were seeded in duplicate (5 × 10^4^ in 400µL of endothelial cell medium) in a 24-well plate.

The production of ROS (mainly O_2_^−^) by THP-1 foam cells exposed to HT was evaluated by a NBT (nitroblue tetrazolium) assay, performed as previously described [58]. Briefly, to each well of a 96-well plate were added: 100 μL potassium phosphate buffer (pH 7.8, 50 mM), catalase (5 μL), NBT (25 μM, 5.6 × 10^−9^ M), xanthine (50 μL, 0.1 mM), xanthine oxidase (50 μL, 0.1 mM) and HT. After the addition of NBT, the plates were incubated at room temperature (1 h). Finally, the absorbance was assessed at 620 nm using a microplate reader (Bio-Rad Laboratories, Inc., Hercules, CA, USA). Enzyme activity (O2- production) was reported as the NBT reduction stimulation index (SI) which was calculated as the optical density (OD) ratio of treated to control cells. The SI for the control was taken to be 1.

### 4.4. Foam Cell Formation Assay and Lipid Content Detection

To assess the lipid content of the foam cells, an Oil Red O staining assay was used. Cells were fixed in formalin for 30 min (10%), then washed twice with PBS and stained with Oil Red O solution (Sigma-Aldrich) for 15 min. After washing with PBS, foam cells were observed with a ZEISS Axioskop 40 (CarlZeiss) light microscope equipped with a Coolsnap VideoCamera. Metamorph (Molecular Devices) software was used for acquiring computerized images. For quantitative analysis, staining with Oil Red O of the macrophage foam cells was extracted with 60% isopropanol to measure the absorbance at a wavelength of 492 nm. The number of foam cells formed in each condition was calculated in triplicate manually and presented as percentage of total cells.

### 4.5. Cholesterol Quantitation Assay

To explore the effect of HT on cholesterol accumulation, THP-1 foam cells were treated with HT (50 μM). HPLC was performed to measure the total cholesterol (TC) or free cholesterol (FC) concentration of samples. Foam cells were processed in an ice bath in 1 mL 0.9% NaCl with an ultrasonic homogenizer (Sonopuls) and then centrifugated (12.000 rpm, 7 min). The supernatants were then collected and vortexed. Cholesterol was extracted using isopropanol/hexane (2:3, *v*/*v*) and dissolved in isopropanol (50 mg/mL). The cholesterol standard curve (0, 20, 40, 60, 80 and 100 μg/mL) was generated using standard cholesterol solutions. For the analysis, 50 μL of cell samples and of diluted standards were incubated for 60 min at 37 °C with 50 μL of reaction mixture (500 mM MgCl2, 500 mM TriS-HCl (PH = 7.4), 10 mM dithiothreitol and 5% NaCl). To detect TC content, 0.4 U cholesterol oxidase combined with 0.4 U cholesterol esterase was added. In contrast, the FC content was assessed without adding cholesterol esterase. After incubating, the supernatant was analyzed by a high-performance liquid chromatographry (Agilent Technologies, CA, USA). Isopropanol: heptane: acetonitrile (35 : 12 : 53) were used to elute the column at a flow rate of 1 mL/min for 8 min. Data analyses were performed using TotalChrom software (PerkinElmer, Waltham, Massachusetts, United States). The total protein was detected by the BCA method, and the final cholesterol content was determined using the ratio between the cholesterol concentration and the corresponding protein concentration (μg/mg protein). To detect the total cholesterol content, cholesterol esterase was used to hydrolyze cholesterol ester to free cholesterol. The content of cholesterol ester was equal to the difference between total cholesterol and free cholesterol.

### 4.6. RNA Extraction, Reverse Transcription and Real-Time PCR (qPCR)

Cells were stored in 1 mL QIAzol lysis reagent (Qiagen, Hilden, Germany) and total RNA was isolated according to the manufacturer’s protocol. The concentration of total RNA was assessed with a NanoDrop 2000 UV–Vis spectrophotometer (Thermo Scientific, Waltham, MA, USA), and 1 µg of total RNA was transcribed to cDNA using a QuantiTec Revers Transcription Kit with integrated removal of genomic DNA contamination (Qiagen, Hilden, Germany), according to the manufacturer’s instructions. cDNA was employed for real-time PCR assays, and performed in triplicate using GoTaq qPCR Master Mix (Promega, Madison, WI, USA), as previously described [59]. The following conditions were used: 2 min incubation at 95 °C; 40 cycles consisting of 30 s at 95 °C; 60 °C for 1 min; and 30 s at 68 °C. Specific human primer pairs were used to assess the expression of proinflammatory cytokines and adhesion molecules (Table 1). Relative expression of each gene was normalized by the 18s gene using the ΔCt method, where ΔCt = Ct_(IL-1β,TNFα,VCAM-1,ICAM-1,SR-A1,CD36,ABCA1,ABCG1,LOX-1,PPARγ,LXRα)_ − Ct_18s_. Relative fold changes in gene expression were determined by the 2^−ΔΔCt^ method, where ΔΔCt = ΔCt_experimental sample_ − ΔCt_control sample_.

### 4.7. Hematoxylin–Eosin Stain

The hematoxylin–eosin stain assay was performed as described by Sancilio et al. [60]. HUVEC/THP-1 was seeded on glass coverslips; fixed in cold ethanol 80% for 30′ at 4 °C; stained in hematoxylin staining solution for 1 min; washed 4–5 times with tap water; counterstained in alcoholic eosin for 30 s; dehydrated by a series of graded increases in alcohol concentrations (70–90–100%) for 1 min each, cleared in three changes of xylene for 1 min each; and then mounted in Bio Mount (Bio-Optica, Milano, Italy). All the samples were observed under a ZEISS Axioskop 40 light microscope (Carl Zeiss) equipped with a CoolSNAP video camera (Photometrics) supported by Metamorph (Molecular Devices) software for acquiring digital images. The number of cells per field (THP-1 cells, attached on the HUVEC monolayer) were analyzed by counting hematoxylin–eosin-stained cells from five randomly selected fields at 10× magnification from three independent experiments.

### 4.8. Western Blot Analysis

Western blot analysis was performed as described previously [61], using the following antibodies against rabbit polyclonal CD36 (Origene, TA500921S; 1:2000): rabbit polyclonal ABCA1 (Santa Cruz Biotecnology, Sc-58219; 1:300), rabbit polyclonal LXRα (Origene, TA805015S; 1:2000), rabbit polyclonal PPARγ (Millipore, ABN1445; 1:500) and mouse monoclonal β-actin (Santa Cruz Biotechnology, Inc., Dallas, TX, USA). The blots were then incubated for 1 h at room temperature with goat anti-mouse secondary antibody (Sc-2005; 1:2000; Santa Cruz Biotechnology) or polyclonal goat anti-rabbit secondary antibody (Sc-66931; 1:5000; Santa Cruz Biotechnology). The nitrocellulose was scanned using a computerized densitometric system (Bio-Rad Gel Doc 1000, Milan, Italy). Protein levels were normalized to the housekeeping proteins β-actin to adjust for variability in protein loading and expressed as a percentage of vehicle control.

### 4.9. Statistical Analysis

All results were expressed as means ± standard deviation. Statistical significance was calculated by one-way analysis of variance (ANOVA) and *p* < 0.05 values were considered statistically significant.

## 5. Conclusions

Several pharmacological effects have been linked to HT consumption, including antioxidant, anti-bacterial, anti-tumor, cardioprotective, hepatoprotective, neuroprotective and anti-inflammatory activities [25]. We have shown that HT could have therapeutic potential for the treatment of atherosclerosis. Atherosclerosis is considered an inflammatory disease, and the vascular endothelium is involved in many of the processes related to the development of cardiovascular diseases. The transformation of macrophages into foam cells induced by excess oxLDL and the triggering of inflammatory molecules are fundamental characteristics of the formation of atherosclerotic lesions. HT induces a decrease in cholesterol absorption at the macrophagic level and, at the endothelial level, a reduction in the synthesis of proinflammatory cytokines (IL-1β and TNF-α) and adhesion molecules (VCAM-1 and ICAM-1). These conditions would seem to have an important role in the negative modulation of the molecular pathways involved in the triggering of the atherosclerotic process. In light of the scientific evidence obtained from our evaluation, we can hypothesize a new therapeutic treatment that comband ines ordinary pharmacological treatment with the use of HT. However, there are some limitations of this study. Here, we focus on in vitro foam cells and, to support our hypothesis and to expand the knowledge on the cellular effects of HT, further studies should investigate the effects of HT on in vivo macrophages of atherosclerosis.

## Figures and Tables

**Figure 1 ijms-24-02057-f001:**
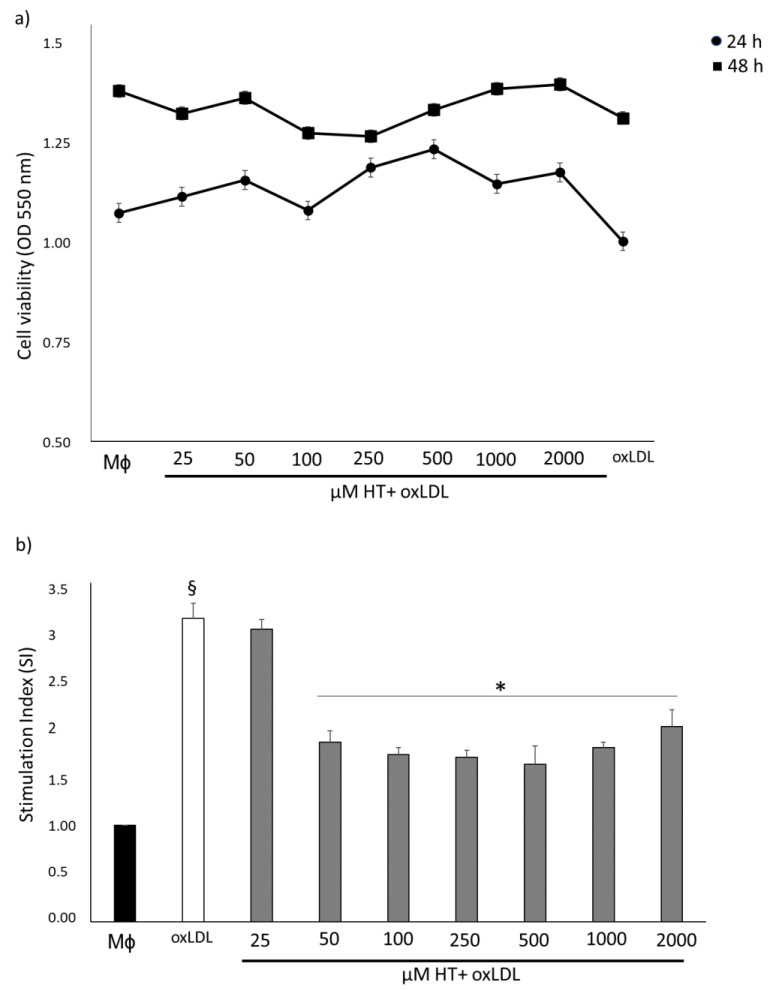
Effect of HT treatment on cell viability and on ROS generation in THP-1 macrophage foam cells. (**a**) THP-1 foam cells were treated with various doses of HT for 24 h and 48 h, and the MTT assay was performed. HT showed no cytotoxicity on foam cells. (**b**) ROS production increased significantly in THP-1 macrophage treated with ox-LDL. Concentrations of HT greater than 50 µM blocked ox-LDL-induced ROS production. The SI for the control was taken to be 1. § *p* < 0.05 vs. Mϕ; * *p* < 0.05 vs. foam cells (*n* = 3). HT, hydroxytyrosol; oxLDL, oxidized low-density lipoprotein.

**Figure 2 ijms-24-02057-f002:**
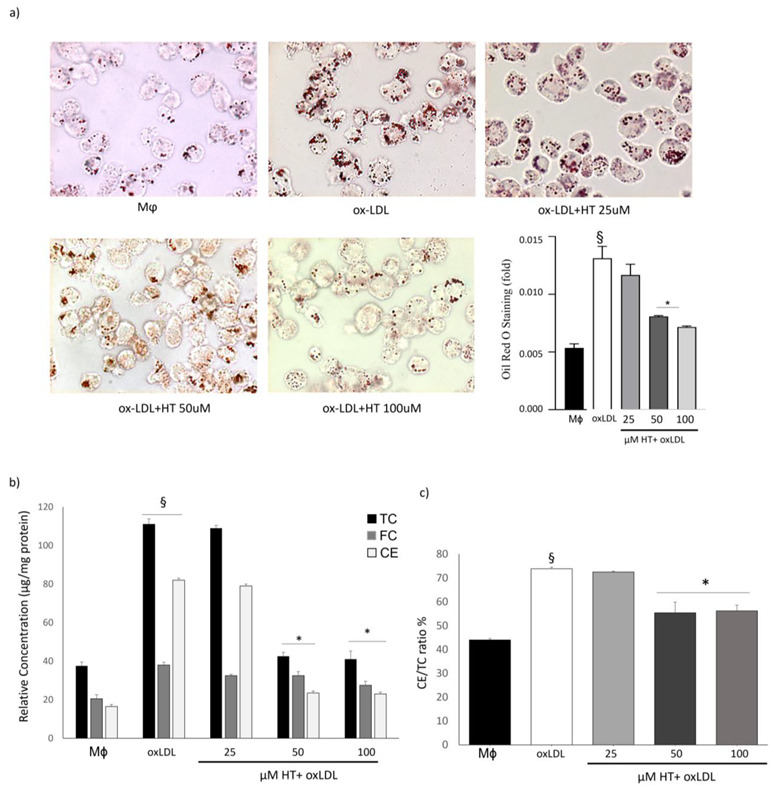
Effects of HT on intracellular lipid accumulation in THP-1 foam cells. (**a**) Foam cells were treated with the indicated concentrations of HT. The cells were stained with Oil Red O (ORO) and evaluated by microscopic (10X magnification) and spectrophotometric analysis. Quantitative assessment of the percentage of lipid accumulation. § *p* < 0.05 vs. macrophages (Mϕ) and * *p* < 0.05 vs. foam cells (*n* = 3). (**b**) The content of intracellular TC, FC and CE. (**c**) The ratio of CE to TC. ox-LDL increased the content of TC, FC and CE and the value of CE/TC. HT decreased TC content in foam cells (*n* = 3). TC, total cholesterol; FC, free cholesterol; CE, cholesterol ester. §: *p* < 0.01 vs. Mϕ; * *p* < 0.05 vs. ox-LDL.

**Figure 3 ijms-24-02057-f003:**
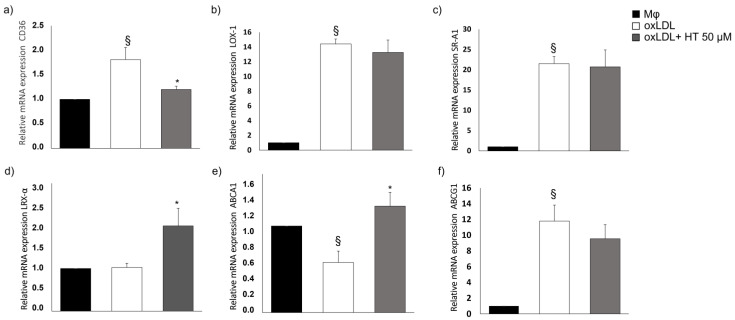
Effect of HT on expression of the major molecules responsible for cholesterol metabolism in foam cells: Gene expression levels of CD36 (**a**), LOX-1 (**b**), SR-A1 (**c**), LXRα (**d**), ABC transporters A1 (**e**) and G1 (**f**) were determined by quantitative RT-PCR in macrophages (Mϕ) and foam cells w/o HT. § *p* < 0.001 vs. Mϕ; * *p* < 0.005 vs. foam cells (*n* = 6).

**Figure 4 ijms-24-02057-f004:**
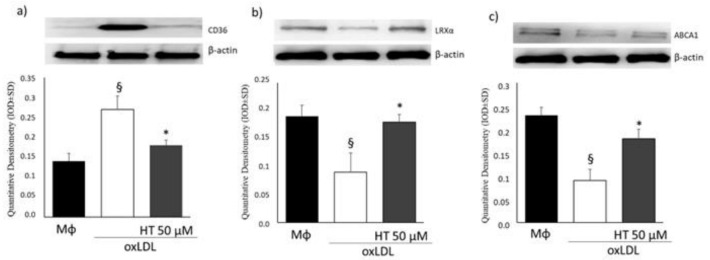
Effect of HT on the expression of CD36, LXRα and ABCA1. The protein levels of CD36 (**a**), LXRα (**b**) and ABCA1 (**c**) were determined using Western blotting and the corresponding antibodies. Β-actin was used as an internal control. § *p* < 0.05 vs. macrophages (Mϕ); * *p* < 0.05 vs. oxLDL (*n* = 3). HT, hydroxytyrosol; oxLDL, oxidized low-density lipoprotein.

**Figure 5 ijms-24-02057-f005:**
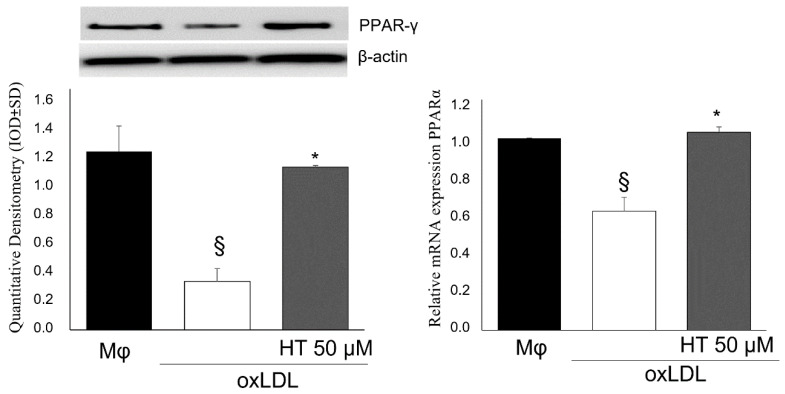
Effect of HT on PPARγ expression in THP-1 macrophage foam cells. THP-1 macrophage- derived foam cells were treated with HT (50 µM) for 24 h. Then, Western blot (**left**) and qPCR (**right**) analyses were performed. § *p* < 0.05 vs. macrophages (Mϕ); * *p* < 0.05 vs. oxLDL (*n* = 3). HT, hydroxytyrosol; oxLDL, oxidized low-density lipoprotein.

**Figure 6 ijms-24-02057-f006:**
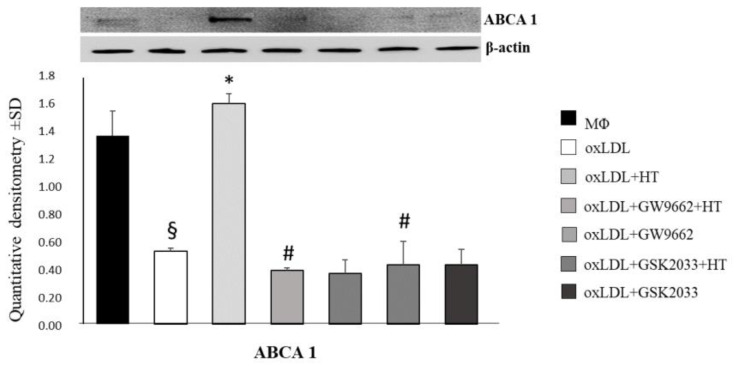
HT influences cholesterol efflux via the PPARγ/LXRα/ABCA1 pathway. ABCA-1 protein expression in foam cells treated with HT alone or in combination with the PPARγ and LXRα antagonists (GW9662 and GSK2033, respectively). Pre-treatment with the antagonists attenuates or abolished the HT-induced upregulation of ABCA-1. § *p* < 0.05 vs. macrophages (Mϕ); * *p* < 0.05 vs. OxLDL; # *p* < 0.001 vs. ox-LDL + HT (*n* = 3). HT, hydroxytyrosol; oxLDL, oxidized low-density lipoprotein.

**Figure 7 ijms-24-02057-f007:**
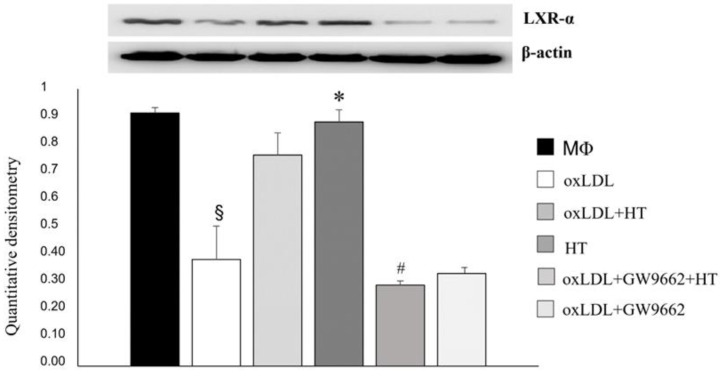
LXRα is involved in the upregulation of ABCA1 induced by HT. THP-1 macrophage- derived foam cells were incubated with GW9662, a PPARγ antagonist, and treated with HT (50 µM). Then, Western blot analyses were performed. § *p* < 0.001 vs. MΦ; * *p* < 0.001 vs. ox-LDL; # *p* < 0.001 vs. ox-LDL + HT (*n* = 3). HT, hydroxytyrosol; oxLDL, oxidized low-density lipoprotein.

**Figure 8 ijms-24-02057-f008:**
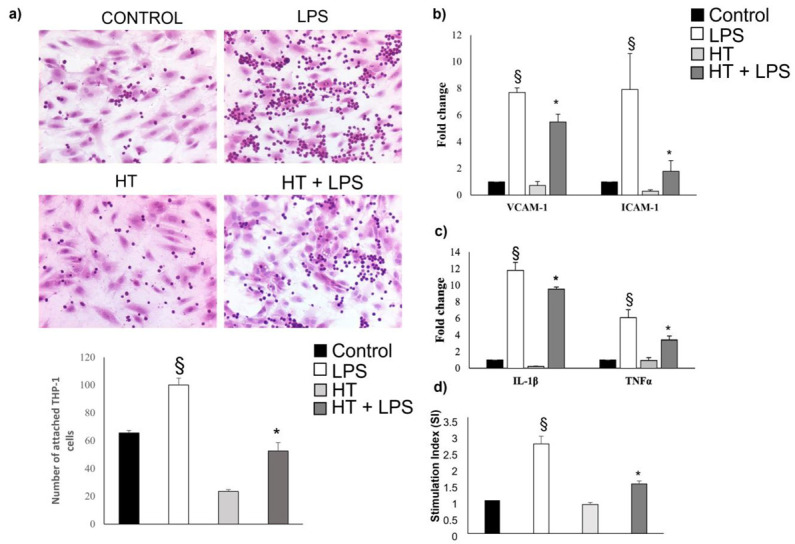
HT decreases LPS-induced adhesion and inflammatory responses in endothelial cells. (**a**) The figure shows the ability of monocytes to adhere to inflamed HUVECs. The administration of HT results in a decrease in monocyte adhesion to inflamed cells. Representative fields of hematoxylin–eosin-stained samples (10X magnification). Expression of (**b**) ICAM-1/VCAM-1 and (**c**) IL-1β/TNF-α was measured by qPCR. (**d**) ROS production decreases significantly in activated HUVEC cells treated with 50 µM HT. All results are expressed as means ± S.E.M. of six independent experiments. § *p* < 0.001 vs. control cells; * *p* < 0.05 vs. LPS-treated cells.

**Table 1 ijms-24-02057-t001:** Primer pair sequences used in the study.

Gene	Forward Sequence (5′-3′)	Reverse Sequence (5′-3′)
IL-1β	TGAGGATGACTTGTTCTTTGAAG	GTGGTGGTCGGAGATTCG
TNFα	CCTTCCTGATCGTGGCAG	GCTTGAGGGTTTGCTACAAC
VCAM-1	GTGGACATAAGAAACTGGAAAAGGG	CATTCACGAGGCCACCACTC
ICAM-1	TGATGGGCAGTCAACAGCTA	GGGTAAGGTTCTTGCCCACT
SR-A1	CTCGTGTTTGCAGTTCTCA	CCATGTTGCTCATGTGTTCC
CD36	CAAGCTCCTTGGCATGGTAGA	TGGATTTGCACAATATGAA
ABCG1	GAAGGTTGCCACAGCTTCTC	CATGGTCTTGGCCAGGTAGT
LOX-1	TTACTCTCCATGGTGGTGCC	AGCTTCTTCTGCTTGTTGCC
PPARγ	GCAGTGGGGATGTCTCATAATGC	CAGGGGGGTGATGTGTTTGAA
LXRα	AAGCCCTGCATGCCTACGT	TGCAGACGCAGTGCAAACA
ABCA1	CCCTGTGGAATGTACCTATGTG	GAGGTGTCCCAAAGATGCAA
18s	CTTTGCCATCACTGCCATTAAG	TCCATCCTTTACATCCTTCTGTC

## Data Availability

Not applicable.

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
