# Peer review of "Hydroxytyrosol Reduces Foam Cell Formation and Endothelial Inflammation Regulating the PPARγ/LXRα/ABCA1 Pathway"

_ijms, 2023, doi:10.3390/ijms24032057_

Round 1
Reviewer 1 Report
Hydroxytyrosol (HT) is a specific polyphenol of olive oil only found in this oil which is associated with the Mediterranean diet.
At the moment, the biological activity of HT is still not well known and therefore this research paper has lot of interest whereas it must be strongly improved.
1) Introduction
It must be written that HT is a major plyphenol of olive oil and that it is characteristic of this oil. No paper on the profil of olive oil and on its content on HT are cited. I suggest to cite the following paper, this is a rare paper which clearly show that HT is only found in olive oil
Zarrouk A, Martine L, Grégoire S, Nury T, Meddeb W, Camus E, Badreddine A, Durand P, Namsi A, Yammine A, Nasser B, Mejri M, Bretillon L, Mackrill JJ, Cherkaoui-Malki M, Hammami M, Lizard G. Profile of Fatty Acids, Tocopherols, Phytosterols and Polyphenols in Mediterranean Oils (Argan Oils, Olive Oils, Milk Thistle Seed Oils and Nigella Seed Oil) and Evaluation of their Antioxidant and Cytoprotective Activities. Curr Pharm Des. 2019;25(15):1791-1805. doi: 10.2174/1381612825666190705192902. PMID: 31298157.
In addition, in the introduction, no paper on the nenefits of Mediterranean diet are cited and nothing is said on polyphenols (benefits on aging, age-related diseases (this include sure cardiovascular diseases but also other diseases). The following paper could be cited
Yammine A, Namsi A, Vervandier-Fasseur D, Mackrill JJ, Lizard G, Latruffe N. Polyphenols of the Mediterranean Diet and Their Metabolites in the Prevention of Colorectal Cancer. Molecules. 2021 Jun 8;26(12):3483. doi: 10.3390/molecules26123483. PMID: 34201125; PMCID: PMC8227701.
2) Results
This part is clear.
However, it is difficult to read the figures and the quality of the figure, especially histological pictures, must be improved.
Under their present form the quality of pictures is average.
2) Discussion
This part is also fine.
It must be mentionned that polyphenols could (may be act on LXR receptors); interaction of polyphenols with receptors is suspected. Could you introduce the following refs and discuss this eventuallity and its consequence?
Namsi A, Nury T, Khan AS, Leprince J, Vaudry D, Caccia C, Leoni V, Atanasov AG, Tonon MC, Masmoudi-Kouki O, Lizard G. Octadecaneuropeptide (ODN) Induces N2a Cells Differentiation through a PKA/PLC/PKC/MEK/ERK-Dependent Pathway: Incidence on Peroxisome, Mitochondria, and Lipid Profiles. Molecules. 2019 Sep 11;24(18):3310. doi: 10.3390/molecules24183310. PMID: 31514417; PMCID: PMC6767053.
Fouache A, Zabaiou N, De Joussineau C, Morel L, Silvente-Poirot S, Namsi A, Lizard G, Poirot M, Makishima M, Baron S, Lobaccaro JA, Trousson A. Flavonoids differentially modulate liver X receptors activity-Structure-function relationship analysis. J Steroid Biochem Mol Biol. 2019 Jun;190:173-182. doi: 10.1016/j.jsbmb.2019.03.028. Epub 2019 Apr 5. PMID: 30959154.
3) A conclusion is required
It must underline the interest of HT in the treatment of cardiovascular diseases and other chronic inflammatory diseases
A new paragraph must be done.
4) Material and Methods
NBT has been used to evaluate oxidative stress. It is good method whereas more interesting methods are available at the moment. I am not sure that NBT is only oxidized by superoxide anion. This what it said, but to my opinion other ROS can be involved. Therefore I think that this must be modified. It must be said that NBT can be oidized by ROS including superoxide anion. Appropriate modifications must be done in the whole manuscript.
Major revision required
Author Response
Chieti, 30.12.2022
Manuscript Revised
Dear Reviewer,
we resubmit our manuscript with the changes made as a result of the requests. Your suggestions allowed us to improve our manuscript " Hydroxytyrosol reduces foam cell formation and endothelial inflammation regulating the PPARγ/LXRα/ABCA1 pathway". Below are the detailed responses to the requested revisions.
Sincerely yours,
Lorenza Speranza
Reviewer 1
Hydroxytyrosol (HT) is a specific polyphenol of olive oil only found in this oil which is associated with the Mediterranean diet.
At the moment, the biological activity of HT is still not well known and therefore this research paper has lot of interest whereas it must be strongly improved.
1) Introduction
It must be written that HT is a major plyphenol of olive oil and that it is characteristic of this oil. No paper on the profil of olive oil and on its content on HT are cited. I suggest to cite the following paper, this is a rare paper which clearly show that HT is only found in olive oil
Zarrouk A, Martine L, Grégoire S, Nury T, Meddeb W, Camus E, Badreddine A, Durand P, Namsi A, Yammine A, Nasser B, Mejri M, Bretillon L, Mackrill JJ, Cherkaoui-Malki M, Hammami M, Lizard G. Profile of Fatty Acids, Tocopherols, Phytosterols and Polyphenols in Mediterranean Oils (Argan Oils, Olive Oils, Milk Thistle Seed Oils and Nigella Seed Oil) and Evaluation of their Antioxidant and Cytoprotective Activities. Curr Pharm Des. 2019;25(15):1791-1805. doi: 10.2174/1381612825666190705192902. PMID: 31298157.
In addition, in the introduction, no paper on the benefits of Mediterranean diet are cited and nothing is said on polyphenols (benefits on aging, age-related diseases (this include sure cardiovascular diseases but also other diseases). The following paper could be cited
Yammine A, Namsi A, Vervandier-Fasseur D, Mackrill JJ, Lizard G, Latruffe N. Polyphenols of the Mediterranean Diet and Their Metabolites in the Prevention of Colorectal Cancer. Molecules. 2021 Jun 8;26(12):3483. doi: 10.3390/molecules26123483. PMID: 34201125; PMCID: PMC8227701.
Dear reviewer, we have expanded the introduction by inserting the characteristics of HT and its benefits in the Mediterranean diet. Thank you for your suggestion
2) Results
This part is clear.
However, it is difficult to read the figures and the quality of the figure, especially histological pictures, must be improved.
Under their present form the quality of pictures is average.
We have tried to improve the quality of the figures by working on the resolution and contrast as required. Unfortunately, in some images derived from western blot analysis, the poor resolution is due to the type of antibody used.
2) Discussion
This part is also fine.
It must be mentionned that polyphenols could (may be act on LXR receptors); interaction of polyphenols with receptors is suspected. Could you introduce the following refs and discuss this eventuallity and its consequence?
Namsi A, Nury T, Khan AS, Leprince J, Vaudry D, Caccia C, Leoni V, Atanasov AG, Tonon MC, Masmoudi-Kouki O, Lizard G. Octadecaneuropeptide (ODN) Induces N2a Cells Differentiation through a PKA/PLC/PKC/MEK/ERK-Dependent Pathway: Incidence on Peroxisome, Mitochondria, and Lipid Profiles. Molecules. 2019 Sep 11;24(18):3310. doi: 10.3390/molecules24183310. PMID: 31514417; PMCID: PMC6767053.
Fouache A, Zabaiou N, De Joussineau C, Morel L, Silvente-Poirot S, Namsi A, Lizard G, Poirot M, Makishima M, Baron S, Lobaccaro JA, Trousson A. Flavonoids differentially modulate liver X receptors activity-Structure-function relationship analysis. J Steroid Biochem Mol Biol. 2019 Jun;190:173-182. doi: 10.1016/j.jsbmb.2019.03.028. Epub 2019 Apr 5. PMID: 30959154.
Dear reviewer, thank you for your suggestions, we have made the changes you requested in the text.
3) A conclusion is required
It must underline the interest of HT in the treatment of cardiovascular diseases and other chronic inflammatory diseases
A new paragraph must be done.
Thanks for the suggestion, we have included a concluding section.
4) Material and Methods
NBT has been used to evaluate oxidative stress. It is good method whereas more interesting methods are available at the moment. I am not sure that NBT is only oxidized by superoxide anion. This what it said, but to my opinion other ROS can be involved. Therefore, I think that this must be modified. It must be said that NBT can be oxidized by ROS including superoxide anion. Appropriate modifications must be done in the whole manuscript.
Dear Reviewer, Thank you for this comment. It is true that NBT mainly reveals the levels of superoxide anion produced but, considering the chemistry of these molecules, we agree that we cannot exclude that other ROS may interfere with the oxidation process. We have corrected all the manuscript.
Finally, we reviewed the entire manuscript for English.
Reviewer 2 Report
The authors investigated the effects of hydroxytyrosol (HT), a phenolic compound with known anti-inflammatory and antioxidant properties similar to polyphenols, on macrophage foam cells in vitro. Using cultured THP-1 macrophages treated with oxidized LDL, they assessed the ability of optimal HT doses to inhibit superoxide production, and intracellular lipid and cholesterol accumulation, and modulate factors involved in cholesterol metabolism. They found that HT favorably affects foam cell metabolism in an anti-atherogenic direction, based on its ability to act as a PPARγ agonist, which in turn activates Liver X Receptor-alpha (LXRα) and ABCA-mediated reverse cholesterol transport. This is a well-done study, which does not really require appreciable revision (except for English usage). I was wondering, though, whether HT is an intermediate in tyrosine or catecholamine metabolism with regulatory functions analogous to the antioxidant properties of various thiols, perhaps as a catecholamine or thyroxine analog. If so, this information should be included in the Introduction and/or Discussion.
Author Response
Gentile Revisore,
riproponiamo il nostro manoscritto con le modifiche apportate a seguito delle richieste. I tuoi suggerimenti ci hanno permesso di migliorare il nostro manoscritto "L'idrossitirosolo riduce la formazione di cellule schiumose e l'infiammazione endoteliale regolando la via PPAR γ /LXR α /ABCA1". Di seguito sono riportate le risposte dettagliate alle revisioni richieste.
Cordiali saluti,
Lorenza Speranza
Revisore 2
Gli autori hanno studiato gli effetti dell'idrossitirosolo (HT), un composto fenolico con note proprietà antinfiammatorie e antiossidanti simili ai polifenoli, sulle cellule schiumose dei macrofagi in vitro. Using cultured THP-1 macrophages treated with oxidized LDL, they assessed the ability of optimal HT doses to inhibit superoxide production, and intracellular lipid and cholesterol accumulation, and modulate factors involved in cholesterol metabolism. They found that HT favorably affects foam cell metabolism in an anti-atherogenic direction, based on its ability to act as a PPARγ agonist, which in turn activates Liver X Receptor-alpha (LXRα) and ABCA-mediated reverse cholesterol transport. This is a well-done study, which does not really require appreciable revision (except for English usage). I was wondering, though, whether HT is an intermediate in tyrosine or catecholamine metabolism with regulatory functions analogous to the antioxidant properties of various thiols, perhaps as a catecholamine or thyroxine analog. If so, this information should be included in the Introduction and/or Discussion.
Caro recensore, grazie per il suggerimento. Abbiamo incluso le informazioni nell'introduzione. Abbiamo anche rivisto l'intero testo per l'inglese.